# Role of Operating Conditions in a Pilot Scale Investigation of Hollow Fiber Forward Osmosis Membrane Modules

**DOI:** 10.3390/membranes9060066

**Published:** 2019-06-03

**Authors:** Victoria Sanahuja-Embuena, Gabriel Khensir, Mohamed Yusuf, Mads Friis Andersen, Xuan Tung Nguyen, Krzysztof Trzaskus, Manuel Pinelo, Claus Helix-Nielsen

**Affiliations:** 1Department of Environmental Engineering, Technical University of Denmark, Miløvej 113, 2800 Kongens Lyngby, Denmark; 2Aquaporin A/S, Nymøllevej 78, 2800 Kongens Lyngby, Denmark; mfr@aquaporin.com (M.F.A.); xtn@aquaporin.com (X.T.N.); ktr@aquaporin.com (K.T.); 3Department of Chemical and Biochemical Engineering, Technical University of Denmark, Søltofts Plads, 2800 Kongens Lyngby, Denmark; gabiih@live.dk (G.K.); Moe0207@hotmail.com (M.Y.); mp@kt.dtu.dk (M.P.)

**Keywords:** hollow fiber, forward osmosis, aquaporin, thin film composite

## Abstract

Although forward osmosis (FO) membranes have shown great promise for many applications, there are few studies attempting to create a systematization of the testing conditions at a pilot scale for FO membrane modules. To address this issue, hollow fiber forward osmosis (HFFO) membrane modules with different performances (water flux and solute rejection) have been investigated at different operating conditions. Various draw and feed flow rates, draw solute types and concentrations, transmembrane pressures, temperatures, and operation modes have been studied using two model feed solutions—deionized water and artificial seawater. The significance of the operational conditions in the FO process was attributed to a dominant role of concentration polarization (CP) effects, where the selected draw solute and draw concentration had the biggest impact on membrane performance due to internal CP. Additionally, the rejection of the HFFO membranes using three model solutes (caffeine, niacin, and urea) were determined under both FO and reverse osmosis (RO) conditions with the same process recovery. FO rejections had an increase of 2% for caffeine, 19% for niacin, and 740% for urea compared to the RO rejections. Overall, this is the first extensive study of commercially available inside-out HFFO membrane modules.

## 1. Introduction

Forward osmosis (FO) is an emerging technology utilizing a gradient of osmotic pressure across the membrane as a driving force for water transport. Pressure-less operation results in a unique process with a low fouling propensity [1], a high rejection of solutes [2], and low energy consumption [3]. Osmotically driven membrane processes have been intensively studied for decades in academia, but it was not until recently that progress in commercialization of this technology has manifested through multiple industrial applications of commercial FO membranes in pilot and full-scale installations (e.g., [4,5,6,7,8,9]). Due to the relatively early stage of this technology’s development, FO membranes with diverse form factors can be found on the market. Thus, Fluid Technology Solutions (FTSH_2_O) and Toray offer spiral wound membrane elements, Porifera provides plate-and-frame modules, and Toyobo and Aquaporin A/S both provide hollow fiber modules. Modules offered by Toyobo are based on outer-selective cellulose acetate fibers, whereas Aquaporin A/S manufactures inner-selective biomimetic thin film composite (TFC) hollow fiber forward osmosis (HFFO) membranes. In applications where two liquids are in contact with a membrane surface, such as dialysis, membrane contactors, or gas separation, hollow fiber configuration has been proven to have multiple advantages compared to other form factors [10,11,12]. Also, in traditional filtration processes such as ultra or microfiltration, hollow fiber modules present a competitive solution [1,10]. The most prominent advantages of the hollow fiber module are their high packing density, self-supporting structure and uniform flow distribution facilitating mass transfer and cleanability of the membrane [1]. This indicates that the hollow fiber configuration for the FO process is very promising from a practical and commercial point of view. 

According to the available literature (e.g., [13,14]), an ideal FO membrane should consist of an ultra-thin and dense selective layer, ensuring high water permeability and solute rejection. The layer should be deposited on a highly porous and thin support, giving sufficient mechanical stability and minimal resistance towards diffusion of the draw solute; thereby allowing maximal utilization of the available osmotic pressure as internal (ICP) and external concentration polarization (ECP) inhibit mass transfer, reducing the available driving force. In addition, high membrane hydrophilicity tends to diminish fouling propensity [15]. Both ICP and ECP are influenced by the properties of the solutes, hydrodynamics of the fluid, and membrane structure and composition [16]. As a result, selection of optimal process parameters [17,18,19], suitable draw solution (DS) [20], and draw recovery strategy [21] are all important to maximize the overall performance of the FO process. To date, assessments of FO membranes have not been standardized and multiple examples of the testing conditions and testing setup configurations have been reported [4,7,17,22,23,24,25,26,27]. Even more challenging is the testing and comparison of FO membranes in commercial modules. The hydrodynamic limitations of individual module geometries might contribute significantly to the reduction of the performance, thereby rendering comparisons with lab scale membrane coupon testing difficult [28,29].

Regardless of the application, the efficiency and attractiveness of the FO process is heavily dependent on the solute rejection and multiple studies have described mechanisms and parameters responsible for solute rejection in the FO operation. For example, Jin et al. [30] examined the rejection of pharmaceutical compounds with a 60 cm^2^ TFC–FO membrane and reported that an increase in water flux by double NaCl DS concentration did not influence forward solute rejection. In contrast, Yamamoto et al. [31] reported an increase in solute rejection, with an increased concentration of MgCl_2_ used as a DS. Also, for challenging compounds, such as urea and boron, FO rejection has been tested. For instance, lab-scale investigations of Shon et al. [32] revealed that using a configuration where the DS was facing the active layer (pressure retarded osmosis (PRO) mode) reduced membrane rejection for boron in comparison to the system with the DS facing the support layer (FO mode). The effect of these configurations (PRO and FO mode) were also found to be critical by Park et al. [7], as ECP becomes a key factor of membrane performance. Park et al. [7] minimized the negative effects of ECP by developing an HFFO membrane with a novel interfacial polymerization (IP) process and by changing the module configuration. The temperature is another extensively studied parameter [24,25,33,34,35,36] that affects membrane rejection and water flux. For example, Xie et al. [24] described how temperature increased the diffusion coefficient of the contaminants, and thus solute rejection decreased with increasing temperature. The variance in FO testing conditions and membrane behavior in all of these studies only increases the necessity for systematizing the influence of the operational conditions on HFFO membrane performance. 

Cath et al. [37] proposed a standard testing method for an osmotically driven membrane processes. However, the proposed testing method and setup are applicable only for flat sheet membrane coupons. Majeed et al. [38] investigated the testing conditions of HFFO membranes with various DS and different configurations (PRO mode vs. FO mode) of the module. They observed that the performance of the investigated HFFO membranes—quantified as water flux, reverse solute flux, and specific reverse solute flux—are strongly dependent on the flow rates of feed solution (FS) and DS. These tests were carried out using lab scale modules with membrane areas <30 cm^2^. The findings from lab-scale systems are difficult to extrapolate to a pilot plant or full-scale systems. Choi et al. [39] recognized this problem and developed a model to predict membrane performance under different operational conditions for large FO plants. But the model results were only compared to experimental data from spiral wound membranes. Another study in a full scale desalination system by Kim et al. [4] described the relevance of the operational conditions on FO processes and a pilot plant design using spiral wound membranes. They found that due to the high flow rates of the pilot plant, co-current operation of FS and DS facilitated fouling of the membrane. It was also concluded that the initial draw flow rate restricted the maximum number of possible modules in a series, as the pressure build-up on the shell side can break the modules. The transition between lab-scale and full-scale is also challenging for hollow fiber modules. For inner-selective hollow fiber membranes, for which the selective layer is deposited after potting of the fiber into the module [40], non-invasive testing of the membrane performance is only feasible with complete and intact modules. Such modules can have from few to several dozen square meters and a high packing density of the fibers affecting flow conditions. Therefore, a design of a suitable testing protocol for commercial scale HFFO modules is necessary to evaluate their performance under realistic conditions. 

In this study, we collected the different operating conditions used in a diverse array of published studies and chose a selection of them to create a guideline for testing HFFO modules. Specifically, we studied the effect of draw and feed flow rates; draw solute and concentration; transmembrane pressure (TMP); temperature; membrane orientation (PRO, FO, counter-current and co-current modes); and solute rejection of target compounds (caffeine, niacin and urea), under FO and reverse osmosis (RO) conditions. These compounds were chosen for their different molecular weights, which facilitated solute rejection analysis. Using those operating conditions, we investigated performance of the commercial inner-selective biomimetic HFFO membrane Aquaporin Inside™ HFFO220 with a surface area of 2.3 m^2^. To show the validity and uniformity of the operating conditions impact on the HFFO membrane modules, Aquaporin Inside™ HFFO220 modules were post-treated to increase their water flux. Following the literature studies [41,42], we used sodium hypochlorite (NaClO) to enhance the water flux of the membrane, and we compared its performance to the standard HFFO220 module. Explicitly, we correlated the difference in the membrane performance to the properties of the selective layer and described parameters allowing control of the FO operation when the HFFO configuration is applied to a pilot or full-scale application.

## 2. Materials and Methods

### 2.1. Materials 

The membrane modules used are commercially available HFFO220 membrane modules provided by Aquaporin A/S (Kongens Lyngby, Denmark), and their specifications can be found in the Appendix A. The hollow fiber membrane modules have an active membrane area of 2.3 m^2^. The HFFO membranes consist of a biomimetic TFC selective layer [43] supported by polysulfone fibers with an inner diameter of 195 µm. NaClO from Sigma Aldrich (Søborg, Denmark) was used to modify commercially available HFFO membranes to increase water flux. The salts used for the DS were NaCl and MgCl_2_·6H_2_O from AzkoNobel A/S (Mariager, Denmark) and MgSO_4_·7H_2_O purchased from Tab-Sol (Straszęcin, Poland)—all were food-grade quality. For rejection tests, urea (≥99%), niacin (≥98%) and caffeine (≥99%) from Sigma Aldrich were used. All the solutions used in the tests were prepared with deionized water (DI) (σ < 10 µS/cm). 

### 2.2. Methods

#### 2.2.1. HFFO Membrane Modification 

HFFO membrane modules were exposed to 20 mg·L^−1^ NaClO solution at pH 10.5. During the exposure, the solution was recirculated for 2 minutes at the lumen side of the fibers. After the exposure, post-treated modules were intensively flushed with DI water for about 1 h. In this study the modified HFFO modules will be referred to as chlorinated membranes and denoted HF–C. They will be compared with unmodified HFFO modules, referred to as original membranes and denoted HF–O. 

#### 2.2.2. Membrane Characterization

The morphology of the selective layer was investigated with scanning electron microscopy (SEM). Membrane samples were extracted from dried modules and coated with gold using a Leica EM ACE600 coater (Leica Microsystems, Wetzlar, Germany). SEM images from the cross-section and inner surface of the membranes were taken using a high-resolution QuantaTM FEG 250 SEM microscope (Thermo Fisher Scientific, Erlangen, Germany). The surface charge of the investigated membranes was measured with the streaming potential technique. To facilitate the analysis of the surface charge and due to the equipment available, dried fibers extracted from the HFFO modules and mini-modules (consisting of 300 fibers) were prepared. The mini-modules were only created and tested for the surface charge analysis and the HFFO modules used to extract the fibers were discarded. The inner surface zeta potential was determined with a SurPASS™ 3 electro kinetic analyzer (Anton Paar GmbH, Ostfildern, Germany). The streaming potential of the mini-modules was assessed by flushing 1 mM of KCl through the lumen of the fibers. Calculations for the zeta potential using the streaming potential data were done using the Smoluchowski equation [44]. The water permeability coefficient (A), solute permeability coefficient (B), and structural parameters (S parameter) for the investigated membranes were estimated using an adapted Bui et al. [45] approach, where various concentrations of NaCl, MgCl_2_·6H_2_O, and MgSO_4_·7H_2_O were used as DS and water flux (J_w_), and the reverse solute flux (J_s_) of the two investigated membrane types was measured and used for the calculations. Detailed explanations of the method used for these calculations can be found in the Appendix A.

#### 2.2.3. FO Tests with HFFO Modules and Performance Evaluation

Two chlorinated membranes (HF–C) and two original membranes (HF–O) modules were individually tested in an FO setup using the Titan OSMO Inspector developed by Convergence Industry B.V. (Enschede, The Netherlands). The schematic flow diagram of the system is shown in Figure 1. The FO system consisted of two diaphragm pumps (model NF 1.6 by KNF Neuberger Inc., Trenton, NJ, USA), each to supply DS and FS solutions to the HFFO module. The flow of the solutions was controlled by two high precision flow meters (IR-OPFlow 100-11, Tecflow International, Velp, The Netherlands). Pressure at the inlets and outlets to the modules were monitored with 4 manometers. The concentrations of the draw solutes in the DS and concentrate solutions (CS) were monitored with two digital conductivity meters (CT) (model LTC 0.35/23 and LGT 1/23, respectively by Sensortechnik Meinsberg, Waldheim, Germany). A digital palette balance (model KFB-TM 1.5T by KERN, Balingen, Germany) was used to measure accumulation of the CS in the tank. Both FS and DS were stored in tanks and their temperature was controlled by heating jackets (IBC-J90 by Frank Berg Supplies, Delft, The Netherlands). At the beginning of each experiment, membrane modules were rinsed with DI water for 5 min through both the shell and the lumen side. The FO setup was then started with the specific set of operational conditions and was running for 45 min since J_w_ and J_s_/J_w_ were stabilized in the system after a few minutes. Thus, data obtained in 45 min was found to be sufficient for the required analysis. After each experiment, the membrane module was cleaned for 45 min with DI water through both the shell and lumen side. 

The FO performance of tested modules was analysed by measuring J_w_, J_s_ and specific reverse solute flux (J_s_/J_w_). The concentrate weight recorded on the balance was used to calculate J_w_ applying Equation (1): (1)$$J_{w}=\frac{Q_{F,\mathrm{in}}-Q_{F,\mathrm{out}}}{S_{A}}$$
where Q_F,in_ and Q_F,out_ represent the volumetric flow rate of the feed and concentrate (feed outflow) respectively of the module and S_A_ represent the surface area of the membrane. J_s_ was measured experimentally using Equation (2): (2)$$J_{s}=\frac{Q_{F,\mathrm{out}}\cdot \sigma \cdot \beta }{S_{A}}$$
where σ is the average conductivity (µS·cm^−1^) of the concentrate and β is a proportionality coefficient describing the relation between conductivity and salt concentration. Here, β was experimentally evaluated for each salt type used as a DS and equals 0.47, 0.43, 0.49 mg·L^−1^ per µS·cm^−1^ for NaCl, MgCl_2_, and MgSO_4_, respectively.

J_s_/J_w_ can be used to describe the efficiency of the FO process, calculated by division of the J_s_ (Equation (1)) by J_w_ (Equation (2)). 

All experiments were carried out in a single-pass mode following the standard conditions listed in Table 1. Each parameter was modified one-at-a-time, while keeping the rest of the parameters constant. Additionally, for tests with feed flow rates, draw flow rates, and TMP and operation modes, two different FSs, varying in osmotic pressure, were selected (Table 1). The standard conditions described in Table 1 were based on the recommendations of the HFFO membrane module manufacturer and J_w_ and J_s_ characteristics of the membranes to avoid damaging the membranes during all operating conditions tested. A detailed list of the parameters modified during the FO performance tests are listed in Table 2. 

For the test with various DSs, four concentrations of each salt type were applied. In order to compare performance of the DS based on the osmotic pressure they generate, various concentrations of the salts were prepared. Table 3 lists the osmotic pressure of the applied DS as a function of the salt concentration. The calculations were performed using the Van’t Hoff equation [46].

For all the experiments and both membranes (HF–C and HF–O), a statistical analysis was performed to confirm significant differences between the membranes and the experiments. The probability value (p-value) was calculated for J_w_ and J_s_/J_w_ and used to confirm the hypothesis explained during examination of the results. Information on the p-values obtained and the statistical analysis used can be found in the Appendix A.

#### 2.2.4. Forward Solute Rejection in FO and Low Pressure Reverse Osmosis (LPRO)

During the FO process, the permeate flow is diluted with the DS. Therefore, rejection calculations of a target compound must take this dilution into account. Following the literature studies [30,49], if the concentration of the target compound is obtained from the draw outlet flow, instead of the permeate flow, the dilution will create an overestimated rejection, which does not represent the real membrane rejection. Hence, using the mass balance of the system, concentration of the target compound in the permeate is calculated using Equation (3): (3)$$c_{\mathrm{permeate}}=\frac{C_{D,\mathrm{out}}\cdot Q_{D,\mathrm{out}}-C_{D,\mathrm{in}}\cdot Q_{D,\mathrm{in}}}{Q_{\mathrm{permeate}}}$$
where $c_{D,\mathrm{out}}$, $c_{D,\mathrm{in}}$ and $c_{\mathrm{permeate}}$ represent the concentrations of the target compound in the draw outflow, draw inflow, and permeate stream. Q_D,out_, Q_D,in_ and Q_permeate_ stand for draw outflow, draw inflow and permeate volumetric flow rate. Subsequently, membrane rejection can be calculated according to Equation (4): (4)$$R_{\mathrm{FO}}=\left(1-\frac{C_{\mathrm{permeate}}\cdot Q_{\mathrm{permeate}}}{C_{F,\mathrm{in}}\cdot Q_{F,\mathrm{in}}}\right)\cdot 100\%$$
where R_FO_ represents the membrane rejection in the FO process. A solution of 200 mg·L^−1^ of caffeine, niacin and urea were used as FS to analyse membrane rejection in both FO and LPRO processes. Table 4 summarizes the testing conditions applied in these experiments. Lastly, tests to investigate correlation between recovery and urea rejection were carried out. Different recoveries were obtained by changing the feed flow rate in the FO set up for the original and HF–C modules (Table 5). 

Niacin and caffeine concentration analysis used in membrane rejection experiments was carried out with a Genesys™ 10S UV-Vis spectrophotometer (Thermo Fisher Scientific, Erlangen, Germany). Both compounds were analysed at 240 nm. Urea analysis was done by the Urea/Ammonia Assay Kit from Megazyme (Bré, Ireland). This kit uses urease enzyme to degrade urea into ammonia, which is later reacted by glutamate dehydrogenase to produce a fluorescent indicator. Results were analysed with the UV-Vis spectrophotometer.

HF–C and HF–O modules were also tested in LPRO process for rejection of caffeine, niacin, and urea. The schematic diagram of the setup is shown in Figure 2. The LPRO system consisted of a gear pump (model GC-M25.JVS.6 by Micropump, Vancouver, WA, USA) suppling the FS, an analogue flow meter, two analogue manometers situated at the inlet and outlet of the modules, and a needle valve for pressure adjustment. The temperature of the FS was controlled and maintained at 25 °C by a heat exchanger immersed in the feed tank.

Similarly, during LPRO tests, FS contained 200 mg·L^−1^ of the niacin, caffeine, or urea. To obtain the same process recovery (23 ± 4%) for both types of studied membrane modules, flow rates of the FS were set to 1 L·min^−1^ and 0.5 L·min^−1^ for HF–C and HF–O modules, respectively. For all tests, TMP applied was established at 2 bar, calculated according to Equation (5):(5)$$\mathrm{TMP}=\frac{P_{\mathrm{in}}+P_{\mathrm{out}}}{2}$$
where P_in_ and P_out_ are the pressures at the inlet and at the outlet of the HFFO modules.

After approximately one hour of equilibration, tests were carried out and permeate samples were collected for permeability and rejection measurements. The water permeability coefficient was calculated according to Equation (6):(6)$$A=\frac{V}{S_{A}\cdot \mathrm{TMP}\cdot \Delta t}$$
where V is volume of the collected permeate sample and Δt is sample collection time. The collected permeate samples were used for concentration analysis with the UV-Vis spectrophotometer or urea kit analysis mentioned previously. Rejection of the niacin, caffeine, or urea were calculated with Equation (7): (7)$$R_{\mathrm{RO}}=1-\frac{C_{\mathrm{permeate}}}{C_{F,\mathrm{in}}}$$
where R_RO_ is the solute rejection in the RO process.

For all the experiments, rejection (J_w_ and J_s_/J_w_) was calculated from the individual tests and later averaged for HF–O and HF–C modules. Standard deviations were calculated from these values obtained from duplicates for each test.

## 3. Results and Discussion

In this section, characterization and FO performance tests for two types of HFFO membranes are presented and discussed. The analysis of the HFFO membrane provides an insight into the differences in polyamide layer structure between the HF–C and HF–O membranes. Awareness of the difference in the membrane characteristics helps to understand membrane response to the operating conditions and the outcome in FO performance.

### 3.1. Membrane Characterization

The two types of HFFO modules were analyzed to understand the impact of the chlorination treatment on the polyamide layer. Specifically, the difference in surface charge and surface morphology between HF–C and HF–O membranes are reported here. Furthermore, the results of the mathematical model described in the Appendix A, used to estimate the intrinsic characteristics of the FO membranes (A, B, and S parameter) for three different draw solutes, are discussed in this section. 

#### 3.1.1. Zeta Potential

Electrostatic interactions between membrane surface and solute contribute to the solute’s rejection. Zeta potential measurements are the most common technique for evaluating the surface charge of the membrane. In Figure 3, the inner surface zeta potentials as a function of pH for both HF–C and HF–O membranes are shown.

For both membrane types, zeta potential is reduced with a reduction of pH. The isoelectric point was measured at 3.7 and 2.8 for HF–O and HF–C membranes, respectively. The isoelectric point around pH 2–4 was expected due to the nature of the polyamide and the presence of the carboxylic groups at the surface of the polyamide [50]. The shift in the isoelectric point after membrane chlorination could be explained as a side effect of the chlorination post-treatment. The chlorination-promoted hydrolysis of the polyamide C–N bond and led to the cleavage of the polyamide layer. Consequently, the hydrolysis resulted in a rise of the carboxyl group number at the surface of the selective layer, and thus, an increase in the negative surface charge manifested by a more negative zeta potential. 

#### 3.1.2. Scanning Electron Microscopy

The morphology of the selective layer for both HF–C and HF–O membranes was studied with SEM imaging. SEM images of the inner surface and of the fibers cross-section are shown in Figure 4. 

Based on the cross-section images (Figure 4c,d), the thickness of the polyamide layer was estimated to be 70.8 ± 9.8 nm for HF–O membranes and 49 ± 4.9 nm for the HF–C membranes. In contrast, the inner surfaces of the investigated membranes (Figure 4a,b) did not exhibit any significant difference between each other. Chlorine usage is a common method for disinfection in water treatment as it inhibits the growth of harmful microorganisms [51]. However, it is widely known that polyamide membranes are highly sensitive to chlorine solutions, and Simon et al. [52] highlighted the need for chlorine-resistant membranes for water purification processes. The difference in thickness between the membranes can easily be explained by the chlorine attack of NaClO to the polyamide. Under alkaline conditions, two competing mechanisms might occur in the presence of a chlorine species (Figure 5). 

The first mechanism postulated by Orton et al. [53] is known as N-chlorination. Here, the chlorine radicals attack the amidic nitrogen in the polyamide structure, which has a lone electron pair and is prone to share it with the partially positive chlorine radicals. This reaction is reversible and can be further aided in acidic conditions where a direct aromatic substitution will occur, producing a chlorinated aromatic ring (Orton arrangement). Once conditions are alkaline, N-chlorination results in the hydrolysis of the polyamide are promoted by the chlorine attack. In that case, after N-chlorination, polarization of the C–N bond facilitates its hydrolysis by the OH^−^ groups present in the elevated pH of the NaClO solution. This results in the cleavage of the polyamide. As schematically described in Figure 5, the cleavage might result in the reduction of cross-linking and an increase in the number of hydroxy groups in the polyamide structure, which enhances the hydrophilicity of the selective layer. Thus, in accordance with the literature [41,42,52] and our observations, HF–C membranes exhibit reduced polyamide layer thickness, an increase in water permeability, and a reduction in solute rejection. As described by Verbeke et al. [42], severe chlorine-promoted hydrolysis of the polyamide will result in dissolution or separation of the selective layer from the support layer, thereby damaging the membrane. Since the HF–C membranes did not exhibit a drastic decrease in solute retention, and Figure 4a,b showed no difference in membrane surface morphology, we concluded that 20 ppm NaClO solutions can be effectively used to improve J_w_ for FO membranes. 

#### 3.1.3. Evaluation of the A, B, and S Parameters

LPRO tests with HF–O and HF–C were carried out at 2 bars of TMP and 500 mg·L^−1^ NaCl solution as a feed. The LPRO tests resulted in A values of 2.46 ± 0.12 L·m^−^^2^·h^−^^1^·bar^−^^1^ and 1.59 ± 0.13 L·m^−^^2^·h^−^^1^·bar^−^^1^ for HF–C and HF–O modules, respectively. Calculated B parameter was measured as 0.18 ± 0.03 L·m^−^^2^·h^−^^1^ and 0.15 ± 0.01 L·m^−^^2^·h^−^^1^ for HF–C and HF–O modules, respectively. The LPRO findings agree with our expectations, and hollow fibers after chlorination indeed exhibit higher water permeability and higher solute (NaCl) permeability. In order to estimate the A, B, and S parameters in the FO process, the modified Bui et al. [45] model for randomly-packed bundle of fibers was applied. Results obtained from FO tests with 4 different concentrations of NaCl, MgCl_2_, and MgSO_4_ as draw solutions were used as inputs for estimation of the A, B, and S parameters. The summary of the results generated is listed in Table 6. Please note that the applied method is quite sensitive to the quality of the experimental data; experimental errors in J_w_ and J_s_ values may lead to calculations of A, B and S parameters that are not feasible. For example, the tests with the HF–C module and NaCl as a draw solution obtained an A value of 4.04 L·m^−^^2^·h^−^^1^·bar^−^^1^, which is far from the results generated with the same membrane for tests with MgCl_2_ and LPRO. Therefore, the A value was fixed to a more feasible 2.46 L·m^−^^2^·h^−^^1^·bar^−^^1^ (LPRO results), and the calculation was run again. The calculation resulted in a slightly higher root mean square error (RMSE) value (from 0.77 to 1.22), a slightly lower R^2^-J_w_ (from 0.98 to 0.92) and significantly better R^2^-J_s_ (from 0.79 to 0.95). The fact that the RMSE was reduced and R^2^-J_s_ was improved indicates that the method requires subjective judgment and the calculated values should be treated as estimations. The same logic was behind the adjustment of the applied calculation constraint for experiments, where HF–C modules were tested with MgSO_4_. Here, the poor fit was related to the barely measurable reverse solute flux. 

Nevertheless, after subjective treatment of the generated data, we can conclude that the A value ranges from 1.56 L·m^−2^·h^−1^·bar^−1^ to 1.86 L·m^−2^·h^−1^·bar^−1^ for the HF–O module and from 2.46 L·m^−2^·h^−1^·bar^−1^ to 2.74 L·m^−2^·h^−1^·bar^−1^ for HF–C modules. As expected, the values are close to the results obtained for the LPRO test and an increase of water permeability with chlorination is clear for both tests. The S parameter estimation resulted in values of 0.09 mm to 0.18 mm for all the membranes. The obtained results are in agreement with the S parameter reported for Aquaporin Inside^TM^ membranes by Xia et al. [54]. The resulting low S parameter contributes significantly to reduction of the ICP and better utilization of the available driving force. Finally, as expected, the estimated B value for NaCl was the highest from all the draw solutions (0.24 L·m^−2^·h^−1^ and 0.80 L·m^−2^·h^−1^ for HF–O and HF–C, respectively) while the lowest was from MgSO_4_ (0.01 L·m^−2^·h^−1^ and 0.12 L·m^−2^·h^−1^ for HF–O and HF–C, respectively).

### 3.2. FO Tests with Hollow Fiber Modules

The measured J_w_ and J_s_/J_w_ of the HF–O and HF–C modules were compared under different feed and draw flow rates, TMP, draw solutes, operational modes (PRO vs. FO and co-current vs. countercurrent), and temperature. In addition, the effect of two types of FS (DI water and seawater solution) were also compared. Tests with an artificial seawater solution were analyzed using only J_w_ due to the limitations in accurate measurements of the solute permeation from the draw to the feed solution. Moreover, rejection of the model compounds (caffeine, niacin, and urea) with different molecular weights was analyzed when modules were operated in an FO or LPRO setup. The relation between membrane rejection and process recovery was studied using the example of urea as a feed contaminant. 

#### 3.2.1. Role of the Feed Flow Rate

Feed flow rate was varied to investigate the influence of the ECP on the active layer side of the membranes. Here, the feed flow rate varied from 60 to 140 L·h^−1^, and 1 M NaCl was employed as a DS and operated at 25 L·h^−1^. The results for both membrane types are summarized in Figure 6. 

For DI water as FS, the HF–O module exhibited an J_w_ increase from 17.4 to 20.2 L·m^−2^·h^−1^ (about 16.2%) in comparison to HF–C module that showed J_w_ to increase from 20.5 to 26 L·m^−2^·h^−1^ (about 26.7%) by varying feed flow rate in the tested range. As shown in Figure 6a, the J_w_ of the HF–C membranes is higher than that of the HF–O membranes at any testing conditions. J_s_/J_w_ is also increased with feed flow rate for the HF–C membranes but remains relatively constant for HF–O membranes for all tested feed flow rates. Furthermore, J_s_/J_w_ is much lower for HF–O membranes (J_s_/J_w_ = 0.15 ± 0.02 g/L) than it is for HF–C membranes (J_s_/J_w_ = 0.4 ± 0.1 g/L).

An increase of J_w_ for the HF–C membranes was expected. As described in the literature [42] and in accordance with Section 3.1.2 results (Figure 4), chlorination of the TFC polyamide-based layer results in the reduction of the cross-linking density and thickness of the selective polyamide layer. Thus, chlorine treated membranes are more permeable and less selective. The main reason for the higher J_s_/J_w_ of the HF–C membranes is the increase of the salt permeation coefficient B caused by this decrease in solute retention of the polyamide (see Section 3.1.3). 

An increase of the J_w_ with feed flow rate was quite surprising since we do not expect a contribution of the ECP on the active layer side for DI water that is free of any solutes. Nevertheless, the results summarized in Figure 6a clearly indicate that J_w_ for both membranes increases with feed flow rate. Such behavior can only be attributed to the reduction of the ECP by reduction of the mass transfer coefficient with the flow rate. We speculate that here, by introduction of higher feed velocity, a stagnate layer of the fluid at the active layer gets disturbed, and diffused salt from the DS can be more effectively transported to the bulk of the FS. Therefore, the net driving force—the difference in osmotic pressure between the feed side and draw side of active layer—is enhanced. A similar rationalization can be applied to the explanation of the difference in development of J_s_/J_w_ for the HF–C and HF–O membranes with feed flow rate. As such, the low retention HF–C membranes can transport more salt to the feed side, locally increasing its concentration at the feed side of the selective layer. By facilitating the transport of this accumulated salt to the bulk solution via a higher feed flow rate, the driving force for salt transport from DS to the FS is also enhanced. As a result, we can observe an increase in J_s_/J_w_ with feed flow rate for low retention HF–C membranes. In contrast, J_s_/J_w_ for HF–O membranes that exhibit higher retention for salt is not affected by the feed flow rate since salt transport from the draw to the feed side is negligible. We think that this effect is valid for all types of FO membranes, and DS and can be mitigated by selecting a larger molecular size of draw solutes or more selective membranes. This effect also confirms the postulate by Werber et al. [13] that more selective membranes, not more water permeable FOs, are critically needed. 

In these experiments, DI water represents the ideal FS (no osmotic pressure, no solutes), whereas artificial seawater represents the problematic FS (high osmotic pressure, high concentration of solutes). Hence, the use of these two FSs is able to display an overview of the effect of the FS osmotic pressure and its impact on the membrane performance. In the case of artificial seawater as an FS, the obtained J_w_ values are much lower than those obtained for the DI water case (Figure 6b). The lower J_w_ was expected as the osmotic pressure of the feed is significantly higher, and the difference in osmotic pressure between FS and DS is only ΔΠ = 20.5 bar. The J_w_ for HF–O membranes increased from 2.6 ± 0.1 L·m^−2^·h^−1^ to 3.5 ± 0.3 L·m^−2^·h^−1^ and for HF–C membranes from 2.0 ± 0.1 L·m^−2^·h^−1^ to 3.8 ± 0.1 L·m^−2^·h^−1^. Interestingly, in the conditions where the FS exhibits high osmotic pressure, the difference in J_w_ between HF–C and HF–O membranes is diminished. Furthermore, similar to the DI water experiments, an increase in the feed flow rate resulted in an increase in flux, proving the dominance of ECP at the active side of the membrane.

#### 3.2.2. Role of the Draw Flow Rate

Similar to the feed flow rate, the applied draw flow rate has also influenced the performance of HFFO modules. FO performance obtained at various draw flow rates for the two types of investigated HFFO modules is summarized in Figure 7. As listed in Table 1, the feed flow rate was kept constant at 100 L·h^−1^ whereas the flow rate of 1M NaCl DS was varied from 25 to 100 L·h^−1^. 

When DI water was applied as an FS, the HF–O modules exhibited J_w_ of 19.0 ± 1.0 L·m^−2^·h^−1^ at 25 L·h^−1^ and J_w_ systematically increased with the draw flow rate to 25.4 ± 1.0 L·m^−2^·h^−1^ at 100 L·h^−1^ (J_w_ increase of 33.5%). For the HF–C membrane, J_w_ was higher than for the HF–O membrane at any draw flow rate investigated, and it increased from 24.9 ± 0.6 L·m^−2^·h^−1^ at 25 L·h^−1^ to 31.3 L·m^−2^·h^−1^ at 100 L·h^−1^, respectively (J_w_ increase of 25.4%). The rate of J_w_ increase with the draw flow rate does not have a linear character, as it was for the variations of the feed flow rate (Figure 6a), and reaches a plateau value (Figure 7a). 

The larger impact of the draw flow rate on the change in J_w_ in comparison to the feed flow rate can be explained by the more severe effect of the draw flow rate on the dilutive ICP and the dilutive ECP at the shell side of the membrane. The cause of this is simple—dilution of the DS with permeated water contributes significantly to the reduction of the available driving force along the length of the HFFO module. As a result, the average concentration of the DS inside the module is no longer 1 M NaCl and it depends heavily on the draw flow rate and water permeation. The higher the draw flow rate, the less dilution is obtained in the module. In an ideal situation, the DS should be operated at very high flow rates. However, the hammer effect of the liquid entering the module and the pressure build-up at the shell side might cause mechanical damage of the fibers. Therefore, to extend the lifetime of the membrane, low or moderate draw flow rates are more practical. When the draw flow rate increases, an average concentration of draw solute increases because of the shorter residence time of the DS in the module. Moreover, an increase of the mass transfer coefficient makes the boundary layer at the support-bulk interface thinner, allowing accelerated diffusion of the salt from the outer surface of the fiber into the support. Subsequently, dilutive ECP is decreased. An increase in the solute concentration inside the module elevates concentration of the solute in the support, alleviating the detrimental effect of the ICP. Consequently, J_w_ increases as the net osmotic pressure difference across the active layer is higher. The maximum concentration in the module that can be reached is the initial concentration of the DS, and by increasing draw flow rate, we asymptotically approach this concentration. Therefore, the plateau in J_w_ can be observed at higher draw flow rates. Hence, there is an optimal draw flow rate for each HFFO module, above which an increase in J_w_ does not occur, and operation above this draw flow rate makes no sense from a practical and economical point of view. 

In the case of artificial seawater as an FS, a maximum J_w_ obtained was 5.7 L·m^−2^·h^−1^. Herein, for tests with various draw flow rates, no difference between HF–C and HF–O modules was observed in terms of J_w_. The unfavorable conditions (concentration polarization effects and low permeate flux) produced by a high osmotic pressure FS continues to cancel all differences in J_w_ between HF–C and HF–O. Yet, by increasing the draw flow rate, the average concentration of the DS increased and the net driving force for water transport was enhanced. Therefore, by reducing the ICP effect, the J_w_ showed a greater increase (Figure 7b) than in the case of the experiments with various feed flow rates (Figure 6b).

#### 3.2.3. Applied TMP 

FO is a process driven by the difference in osmotic pressure between DS and FS, and, as such, there is no need to apply pressure to obtain water permeation. Additionally, we would not expect a great impact of the TMP on the performance of the HFFO membranes since the hydraulic pressure applied is much lower than the osmotic pressure difference applied in the FO process. Nevertheless, multiple studies have reported both negative and positive impacts of the applied TMP in FO processes [23,38,55,56]. In those studies, elevated TMP was used mostly to study fouling; to facilitate permeate flow rate; or to normalize the draw and feed flow rates, when there are several modules connected in series. In this study, the effect of elevated TMP has been analyzed using the HF–C and HF–O modules. FO performance of the investigated modules at systematically elevated TMP are summarized in Figure 8. For all the testing conditions, where TMP varied from 0.1 to 1.1 bar, the feed and draw flow rates were kept constant and were 100 L·h^−1^ and 25 L·h^−1^, respectively. 

In this range of tested TMPs, J_w_ increased from 17.9 ± 0.5 L·m^−2^·h^−1^ to 19.2 ± 0.5 L·m^−2^·h^−1^ (7.3% increase in J_w_) for HF–O modules and from 24.0 ± 1.3 L·m^−2^·h^−1^ to 27.7 ± 0.2 L·m^−2^·h^−1^ (15% increase in J_w_) for HF–C modules. The results obtained are similar to the ones reported by Coday et al. [23], where minimal changes in J_w_ and J_s_/J_w_ were obtained at elevated TMPs between 1 to 2 bar.

On one hand, the obtained results are expected since TMP does not contribute to a significant improvement of J_w_ for both membrane types. On the other hand, the applied pressure range is quite narrow and performance of the tested HFFO modules could be further increased by increasing TMP. Nevertheless, the fact that FO membranes are purposefully designed with thinner and more-fragile support restricts the possibilities to significantly increase TMP. The difference in the rate of the J_w_ increase with the applied TMP between the HF–O and HF–C modules originates from the difference in the A parameter between those two modules (see Section 3.1.3). A more permeable selective layer, like in the case of the HF–C membrane, will always be more responsive to the applied TMP than a membrane with a less permeable selective layer, like an HF–O membrane. 

J_s_/J_w_ was significantly reduced with applied TMP for HF–C membranes (from 0.40 ± 0.02 g·L^−1^ at 0.1 bar to 0.31 ± 0.2 g·L^−1^ at 1.1 bar) and negligibly for HF–O modules (from 0.14 ± 0.02 g·L^−1^ at 0.1 bar to 0.12 ± 0.2 g·L^−1^ at 1.1 bar). We speculate that the greater increase in J_w_ for HF–C membranes caused a more severe dilutive ICP at the draw side of selective layer. As a result, the solute concentration difference between the draw side and feed side of the active layer was reduced; therefore, the driving force for solute diffusion was diminished. Consequently, we observed reduced J_s_/J_w_ with the applied pressure.

The same minor effects of various TMPs applied to J_w_ were observed for artificial seawater as an FS (Figure 8b). As shown in previous experiments, no differences in J_w_ between HF–O and HF–C membranes were observed when using artificial seawater as an FS.

#### 3.2.4. Influence of the Applied DS

DS selection has been abundantly studied [20,31,57,58,59] in order to tune the FO process for a specific application, to reduce the costs of the process, or to obtain a more environmentally friendly technology. Here, three common inorganic salts (NaCl, MgCl_2_, MgSO_4_) at various concentrations were tested as a draw solute. Concentrations of draw solutes were selected to obtain the same osmotic pressure and compare the DS based on the driving forces that they generate. NaCl and MgCl_2_ osmotic pressures varied from 24.7 to 99.0 bar, whereas MgSO_4_ osmotic pressure was investigated from 7.9 to 74.2 bar due to the limits in solubility of the MgSO_4_. For all the tests, DI water was used as an FS. Feed flow rate and draw flow rate were kept constant at 100 L·h^−1^ and 25 L·h^−1^, respectively. The summary of the results is shown in Figure 9. 

Moreover, for all tests with various DS, the J_w_ obtained with HF–C modules was higher here than the J_w_ obtained for HF–O modules for any DS and for all concentrations investigated. Obviously, for each draw solute, J_w_ increases with an increasing DS concentration since the driving force (osmotic pressure difference) is raised across the membrane. As shown in Figure 9a, by applying NaCl concentration varying from 0.5M to 2M (24.7 bar to 99.0 bar), the HF–O membrane exhibited J_w_ increases from 13.8 ± 0.7 L·m^−2^·h^−1^ to 28.3 ± 1.1 L·m^−2^·h^−1^. In comparison, the HF–C membrane showed a J_w_ increase from 18.6 ± 0.6 L·m^−2^·h^−1^ to 30.1 ± 0.5 L·m^−2^·h^−1^. When MgCl_2_ was applied as a draw solute in the concentrations equivalent in osmotic pressure to the 0.5–2 M range of NaCl (24.7 bar to 99.0 bar), The HF–O membrane showed an increase in J_w_ from 12.6 ± 0.9 L·m^−2^·h^−1^ to 22.5 ± 1.0 L·m^−2^·h^−1^, whereas the HF–C membrane showed a J_w_ increase from 15.6 ± 0.9 L·m^−2^·h^−1^ to 28.3 ± 0.5 L·m^−2^·h^−1^. Finally, for MgSO_4_ as a draw solute, the HF–O membrane exhibited a J_w_ increase from 11.8 ± 0.9 L·m^−2^·h^−1^ to 15.0 ± 0.8 L·m^−2^·h^−1^ in comparison to the HF–C membrane that showed a J_w_ increase from 13.8 ± 0.8 L·m^−2^·h^−1^ L to 21.3 ± 0.1 L·m^−2^·h^−1^. Note that for MgSO_4_, the maximum applied osmotic pressure was 74.2 bar. 

The osmotic pressure of the tested DS is comparable, and, therefore, the differences in J_w_ between them is solely due to the intrinsic properties of the salt and the ICP development for each salt type. Based on the results presented in Figure 9 and the available literature [2,60], we can distinguish two regions of the flux increase with osmotic pressure: 1—a linear increase of the J_w_ with osmotic pressure and 2—an asymptotic increase of J_w_ with osmotic pressure. Obviously, the first regime where J_w_ increases linearly with osmotic pressure represents the most efficient operation regime since applied potential in osmotic pressure is directly proportional to the flux obtained. The second regime with asymptotic increase of the flux with osmotic pressure is related to an extensive ICP. In this regime an increase of the osmotic pressure is not directly proportional to the obtained J_w_, and the available driving force is not efficiently utilized to generate J_w_. As shown in Figure 9a, all tests with NaCl show a close to linear correlation between J_w_ and the applied osmotic pressure of the DS. High diffusivity of small NaCl ions (Table 3) and a low S parameter of the support (Table 6) enhance the transport of salt ions into the support, minimizing dilutive ICP. As a result, a high concentration of the NaCl is experienced by the selective layer, and the highest J_w_ values were obtained for both membrane types using this salt. The diffusion coefficient of MgCl_2_ is slightly lower than NaCl (Table 3), and thus non-linear J_w_ behavior was exhibited by the HF–O membranes at the highest salt concentration. HF–C membranes could maintain a linear J_w_ behavior due to their higher A parameter, delaying severe ICP. However, for MgSO_4_ tests, J_w_ displayed a clear non-linear behavior for both membrane types and here we can explain it as a significantly lower diffusion coefficient (Table 3) of the MgSO_4_, compared to the other draw solutes, which makes this system more prone to ICP contribution at high salt concentrations. The diffusion coefficient is not the only driver for the J_w_ differences between the draw solutes, as it is closely dependent on the molecule/ion size of the solutes [20,61]. Na^+^ and Cl^−^ ions are both small enough to easily diffuse through the membrane support without hampering water transport, thus minimizing ICP. However, for MgCl_2_, the fast-diffusive Cl^−^ ions are impaired by the bigger and slower Mg^2+^ ions to maintain electroneutrality. This causes a slight blockage, by the Mg^2+^ ions, of the water transport from the FS to the DS, and an increase of the ICP. These effects are only further aggravated when Mg^2+^ ions are introduced as MgSO_4_ with SO_4_^2^*^−^* ions, which have the biggest ion size from all the draw solutes tested. Consequently, both of these ions, due to their size and low diffusion rate, have more difficulty travelling through the membrane support, lowering the osmotic pressure difference across the selective layer, and obtaining the lowest J_w_ in this test. 

Nevertheless, compared to NaCl, the lower diffusion coefficients and bigger size of MgCl_2_ and MgSO_4_ can be advantageous in terms of obtaining reduced J_s_ and J_s_/J_w_. More severe ICP for MgCl_2_ and MgSO_4_ results in a lower concentration of the draw solutes at the support-selective layer interface, so a lower driving force for solute transport from the feed to the draw solution is generated. Moreover, TFC active layers generally have a higher retention for divalent ions like Mg^2+^ and SO_4_^2^*^−^* in comparison with monovalent ions, such as Na^+^ and Cl^−^. Those two effects summarized are responsible for the reduced J_s_ and J_s_/J_w_ with MgCl_2_ and MgSO_4_. In an extreme case, a combination of high retention HF–O and MgSO_4_ resulted in a barely measurable J_s_/J_w_ (close to 0 g·L^−1^ for all MgSO_4_ concentrations). In comparison, the tests with NaCl as a DS generated the highest J_s_/J_w_ values.

#### 3.2.5. Operation Mode of the Membrane 

In this section, two membrane operation modes were compared. Firstly, we compared the FO performance of the tested modules during operation in an FO mode (FS facing active layer) and a PRO mode (DS facing active layer solution). Secondly, we investigate the impact of the operation in a co-current and counter-current mode on FO performance of the modules. 

(a) Co-current vs. counter current for HFFO modules

In counter-current operations (used in all experiments in this study), FS and DS enter and exit the HFFO module from opposite sides. In the co-current mode, FS and DS enter and exit the module from the same side of the module. The comparison of the counter-current and co-current operation with the HFFO modules was carried out with 1 M NaCl as DS applied at 25 L·h^−1^. DI water and artificial salt solution were used as an FS and operated at 100 L·h^−1^. As shown in Figure 10a, for the DI water as an FS, for both the HF–C and HF–O modules, the differences in performance between operation in a co-current or counter-current mode were not significant. 

HF–O modules exhibited a J_w_ of 18.2 ± 1.1 L·m^−2^·h^−1^ and 19.0 ± 1.4 L·m^−2^·h^−1^ for co-current and counter-current operations, respectively, whereas HF–C membranes generated higher flux for both operation modes: J_w_ of 23.8 ± 0.8 L·m^−2^·h^−1^ and J_w_ of 24.9 ± 0.6 L·m^−2^·h^−1^ for co-current and counter-current modes, respectively. Similarly, no significant change was measured in J_s_/J_w_ when the two operational modes were compared (Figure 10a). HF–O modules obtained a J_s_/J_w_ of 0.14 ± 0.02 g·L^−1^ and 0.12 ± 0.01 g·L^−1^, and HF–C modules achieved a J_s_/J_w_ of 0.36 ± 0.05 g·L^−1^ and 0.35 ± 0.04 g·L^−1^ for counter-current and co-current modes, respectively. As schematically described in Figure 11, for an FS with low osmotic pressure, such as DI water, the osmotic pressure of the FS does not change significantly during the up-concentration inside the HFFO module. A lack of solutes in the FS and a relatively low diffusion of the salt from the DS to the feed side keeps almost constant osmotic pressure on the FS. Thus, for operation in co-current and counter-current modes, the same average driving force is obtained inside the module, and no difference between operation in co-current and counter-current mode can be measured. 

Also negligible is the difference in J_w_ between counter-current and co-current operation when artificial seawater is applied as an FS (Figure 10b). As reported in the literature [17,38], operation with counter-current mode with high osmotic pressure FS should generate higher flux in comparison to operation in a co-current mode. Due to the maintained constant osmotic pressure difference between FS and DS, the available difference in osmotic pressure is used more efficiently than during operation in the co-current mode (Figure 11). However, this effect cannot be seen in our case due to the low permeation in the process (J_w_ < 4 L·m^−2^·h^−1^). We speculate that a higher difference in osmotic pressure or operating conditions that can enhance J_w_ (see Section 3.2.1, Section 3.2.2 and Section 3.2.4) could reveal the advantage of counter-current operation. 

(b) PRO vs. FO for HFFO modules

Operations in the FO mode (used in all the experiments in this study) and PRO mode were compared with the same concentration of DS (1 M NaCl) and the same feed and draw flow rates (100 L·h^−1^ and 25 L·h^−1^, respectively) for both operating modes. As shown in Figure 12a, when DI water was applied as a DS, HF–O modules exhibited a J_w_ of 17.9 ± 0.5 L·m^−2^·h^−1^ in the FO mode, whereas, in the PRO mode, J_w_ increased to 20.9 ± 1.3 L·m^−2^·h^−1^. Similar behavior was observed for HF–C modules. Here, J_w_ increased from 24.1 ± 1.3 L·m^−2^·h^−1^ in the FO mode and to 26.8 ± 1.3 L·m^−2^·h^−1^ in the PRO mode. The obtained results are in good agreement with the available literature [17,38,60], describing that the dilutive ICP of the DS inside the support side (FO mode) causes a larger J_w_ decline than concentrative ICP of the FS inside the support (PRO mode). However, under similar conditions, the obtained J_w_ improvement (13.9 ± 3.8% increase) during operation in PRO mode is not as significant as reported in the literature [38] (60–80% increase in J_w_). We correlate this rather low impact of the FS and DS orientation on the FO performance to the low S parameter (see Section 3.1.3) of the membrane and generally low ICP when operated in FO mode (see Section 3.2.4). 

As shown in Figure 12a, J_s_/J_w_ is reduced when the process is carried out in PRO mode. In our tests, for HF–C membranes J_s_/J_w_ decreased from 0.4 ± 0.02 g·L^−1^ in FO mode and to 0.27 ± 0.02 g·L^−1^ in PRO mode, respectively. For HF–O membranes, this reduction in J_s_/J_w_ was much lower (0.14 ± 0.01 g·L^−1^ in FO mode and 0.11 ± 0.01 g·L^−1^ in PRO mode, respectively). In PRO mode, diffusion of the solute from the active layer to the bulk of the FS is restricted by the support, similar to the limited solute diffusion into the support when operated in FO mode. We speculate that this limited solute diffusion from the support to bulk solution contributes to a reduction of J_s_/J_w_ in PRO mode.

Like the tests with DI water, when artificial seawater was applied as an FS, the use of the PRO mode results in a slight J_w_ increase. J_w_ increased from 3.6 ± 0.4 to 4.9 ± 0.1 L·m^−2^·h^−1^ and from 4.8 ± 0.1 to 5.1 ± 0.2 L·m^−2^·h^−1^ when switching from the FO to PRO mode for HF–O and HF–C modules, respectively. It can be argued that this J_w_ increase is negligible due to an error originating from the accuracy of the applied measuring technique. In fact, it is expected that the advantages of PRO mode are limited for high osmotic pressure solutions. Due to concentrative ICP of the feed solute, the net difference in osmotic pressure between both sides of the active layer is reduced, and, thus, the PRO mode’s advantages are minimized.

#### 3.2.6. Role of the Temperature 

Multiple studies show that an increase in temperature can effectively enhance water transport in membrane processes [24,25,33,34,35,36]. For example, Seker et al. [25] highlighted the importance of the temperature for viscous FS and described how the efficiency of the FO process can be enhanced by an increase of the FS temperature (reduced viscosity) while keeping the DS cold (reduced draw solute diffusion to the FS). In addition, Xie et al. [24] demonstrated that an increase in J_w_ with temperature can come with a negative side effect of a decreased solute rejection of trace organic compounds, due to the increase in solute diffusivity. Therefore, it is crucial to find the right balance between rejection and J_w_ when choosing the process temperature.

Here, the temperature of both DS and FS varied from 15 °C to 45 °C to determine the temperature impact on the performance of HFFO modules. As before, 1 M NaCl was used as the DS, and feed and draw flow rates were kept at 100 L·h^−1^ and 25 L·h^−1^, respectively. The results for both membrane types are summarized in Figure 13. 

In brief, the HF–O module exhibited a J_w_ increase from 12.7 ± 1.0 L·m^−2^·h^−1^ to 22.0 ± 1.0 L·m^−2^·h^−1^, and HF–C modules showed an increase in J_w_ from 16.1 ± 0.2 L·m^−2^·h^−1^ to 25 ± 0.1 L·m^−2^·h^−1^ by increasing the temperature of FS and DS from 15 °C to 45 °C. Interestingly, for both HF–C and HF–O modules, J_w_ increases to the same extent (about 9.0 L·m^−2^·h^−1^) within the tested temperature range (Figure 13). The increase in J_w_ was anticipated, as temperature influences diffusion rates and fluid viscosity. Firstly, higher diffusivity of the solute enhances its transport inside the support, thereby reducing ICP, and consequently generates higher J_w_ [36]. Secondly, liquid viscosity drops with temperature, thereby reducing resistance for solute diffusion, again improving J_w_ by reduction of ICP. 

Enhanced diffusion of the draw solute, and thus reduced of ICP caused by elevated temperature, also results in an increase in J_s_. As shown in Figure 13, J_s_/J_w_ for HF–O slightly increases with the temperature from 0.11 ± 0.01 g·L^−1^ at 15 °C to 0.17 ± 0.01 g·L^−1^ at 45 °C. This result indicates that J_s_ increases faster than J_w_ with the temperature and facilitates solute diffusion that favors salt transport from the draw side to the feed side over water transport from the feed side to the draw side. We speculate that the lower A value of the HF–O membranes, compared to the HF–C membranes, contributes to this effect. For HF–C membranes, J_s_/J_w_ is relatively constant for tested temperatures (J_s_/J_w_ of about 0.32 g·L^−1^), indicating uniform contribution of the temperature to the enhancement of the salt and water transport.

#### 3.2.7. Rejection of Target Compounds

##### Rejection: FO vs. LPRO

The mechanism behind the solute rejection in pressure driven processes (e.g., RO) and osmotically driven processes (e.g., FO) is discussed in multiple studies [49,62,63,64]. This mechanism is described as a combination of interactions based on steric exclusion, electrostatic repulsion, system transmembrane pressure, process recovery, concentration polarization, and the characteristics of the membrane and the solute. In this section, rejection of the model compounds caffeine, niacin, and urea is compared between two processes—RO and FO. These compounds were chosen due to their differences in molecular weight and molecular charge (Figure 14) and their relevance in food and beverage applications. Urea is an abundant and common contaminant in wastewater, and, as a small and neutral molecule (Figure 14), urea is the most interesting compound to examine due to challenges that membrane technology faces in rejection of this compound.

Feed flow rates for LPRO and FO process were varied to obtain uniform recovery of about 23 ± 4% (Table 4). Concentration of the model compounds in the FS were 200 mg·L^−1^ for all tests. As shown in Figure 15, caffeine rejection of 99 ± 1% was obtained in the FO process, whereas roughly 95 ± 1% caffeine rejection was obtained in pressure driven LPRO for both HF–C and HF–O modules. For niacin with a slightly lower molecular weight, also 100 ± 1% rejection was obtained in FO for both types of HFFO modules. However, in LPRO process, we observed a difference in rejection between HF–C and HF–O module. In particular, niacin rejection in LPRO was 77 ± 1% and 90 ± 1% for HF–C and HF–O modules, respectively. Finally, for the low molecular weight urea, calculated rejection in the FO process was 78 ± 1% and 84 ± 1% for HF–C and HF–O membranes, respectively. Again, for LPRO operations, urea rejection was much lower and for HF–C membrane, we did not observe any urea rejection while HF–O modules showed only 19 ± 1% urea rejection. 

In FO, the HF–C and HF–O membrane showed nearly identical rejection values for caffeine and niacin, and similar urea rejection values at low recoveries. The results confirm that HF–C modules are still selective towards compounds present in the FS regardless of higher J_s_/J_w_ values obtained during previous FO tests. When operated in FO, for the largest compounds, such as caffeine (Figure 14), we cannot distinguish any difference in solute retention between two membrane types, but the difference becomes more pronounced with low molecular weight compounds such as urea. In general, rejection in LPRO is lower for the same compounds than in FO. Thus, LPRO tests reveal more differences in membrane retention for the model contaminants between HF–C and HF–O membranes and the effect of the molecular weight and charge becomes more visible (Figure 15). 

Similarly to the reported studies touching on the comparison between FO and RO solute rejections [31,49,64], solute rejections are also higher for FO process. A larger permeate flux in FO operation can be responsible for this effect. However, Xie et al. [65] compare FO and RO operations at the same permeate fluxes and, in some cases, observed elevated rejection of solutes in FO when compared to RO. Xie suggests that this effect is attributed to the delayed forward diffusion of feed solutes through the selective layer pore when operated in FO mode. However, to fully understand this complex topic, more systematic studies on the interactions between membrane, feed solute, and DS are necessary. 

##### Urea Rejection vs. Recovery

Urea rejection was analyzed under different process recoveries in the FO operation to further understand the correlation between recovery and membrane retention. As mentioned before, urea was selected as a model compound due to the challenges it causes in rejection on the membranes. Feed flow rates were varied from 60 to 140 L·h^−1^ to obtain various recoveries in the module, while the draw flow rate was maintained at 25 L·h^−1^, and the DS used was 1M NaCl. Summarized results on the influence of process rejection on urea rejection are shown in Figure 16. At the lowest feed flow rate (60 L·h^−1^), the highest process recovery (74.1%) was achieved by the HF–C module while the HF–O membrane generated 59.9% recovery. With the highest feed flow rate applied (140 Lh^−1^), a higher process recovery was also obtained for the HF–C membranes (34.2%) than for the HF–O module (25.5%). 

Reduction of the rejection with increasing recovery is expected, as high recovery results in an increase of the solute concentration in the module. At the same time, ECP of the feed solute is enhanced by the rising solute concentration but also by the reduced feed flow rate, decreasing the mass transfer coefficient. Combined, these two effects yielded a higher solute diffusion rate than the feed to the draw side and the lower solute rejection. Interestingly, as shown in Figure 16, HF–C membranes seem to be more sensitive than HF–O membranes to increasing recoveries and reduction of the rejection is more severe for those modules. Here, lower retention of the chlorinated polyamide layer is more pronounced at higher process recoveries. The opposite phenomenon is seen with low recoveries, where HF–C and HF–O membranes have similar rejection. 

## 4. Conclusions

In this study, the significance of the operating conditions to the performance of the FO process for two types of HFFO modules, HF–C and HF–O membranes, has been demonstrated by analyzing their behavior under various testing conditions. The following conclusions can be made from these investigations:J_w_ can be enhanced for the HFFO module by chlorination due to a decrease in the thickness of the polyamide layer.Dilutive ICP was found to be one of the most significant parameters responsible for membrane performance as it acutely limits J_w_ when it becomes severe. This confirmed the dominant role of ICP for FO membranes, independent of membrane characteristics. Consequently, draw concentration and draw solute type have the biggest impact on the membrane performance. Additionally, optimal flow rates and osmotic pressure differences between FS and DS reduce the effect of other operating conditions, such as membrane orientation (PRO/FO mode, co-current/counter-current mode).A low osmotic pressure difference between DS and FS drastically reduces any differences between the chlorinated HFFO modules and the original HFFO modules. However, changes in operating conditions still alter membrane performance following the same mechanism explained for the large osmotic pressure difference between DS and FS.The RO process displayed lower solute rejection compared to the FO process. High process recoveries have a detrimental effect on solute rejection, due to an increase of solute concentration inside the module, which facilitates solute diffusion to the permeate.

This work has demonstrated that operational parameters heavily influence membrane performance for HFFO modules. Consequently, our results can serve as a guideline to take maximum advantage of the HFFO membrane modules in pilot plant or full-scale processes. Furthermore, guidelines can aid FO installation design, thus paving the way for implementation of FO in technological applications.

## Figures and Tables

**Figure 1 membranes-09-00066-f001:**
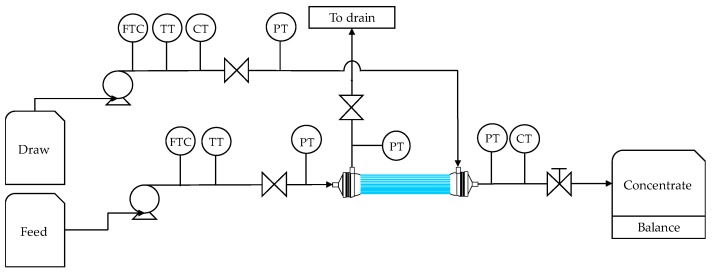
Flow diagram of the Titan OSMO Inspector set up during testing of hollow fiber forward osmosis (HFFO) modules. Flow transmitter controller (FTC), temperature transmitter (TT), conductivity transmitter (CT) and pressure transmitter (PT). Concentrate weight was recorded on a balance whereas diluted draw was discharged.

**Figure 2 membranes-09-00066-f002:**
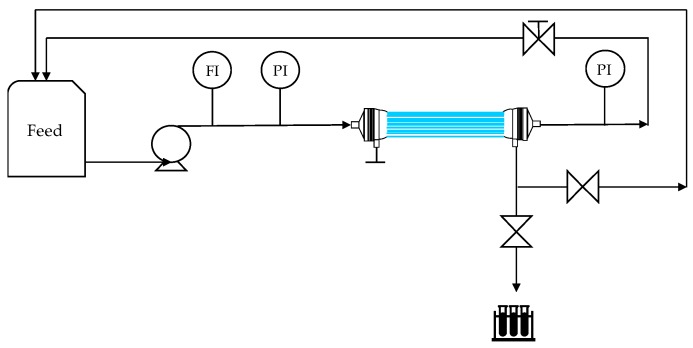
Schematic description of the LPRO setup. Flow meter (FI) and pressure meters (PI).

**Figure 3 membranes-09-00066-f003:**
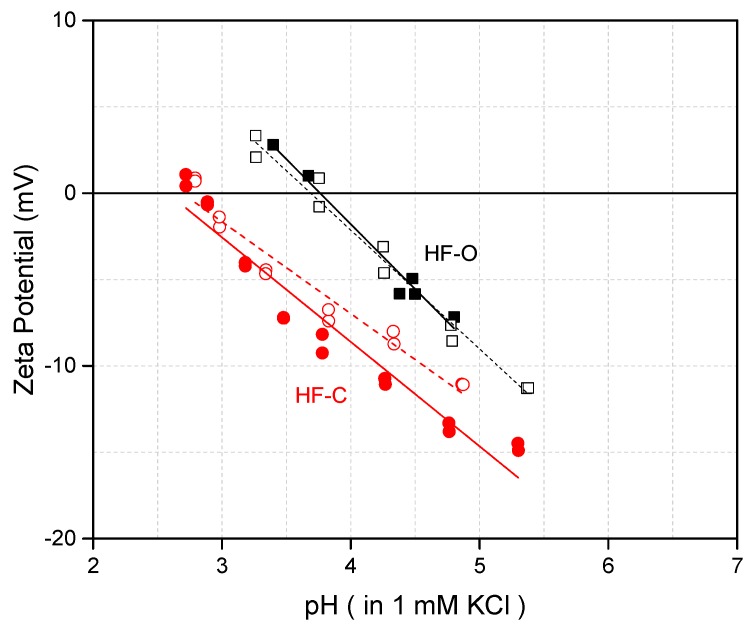
Zeta potential as a function of pH at the inner surface of the HF–C and HF–O membranes.

**Figure 4 membranes-09-00066-f004:**
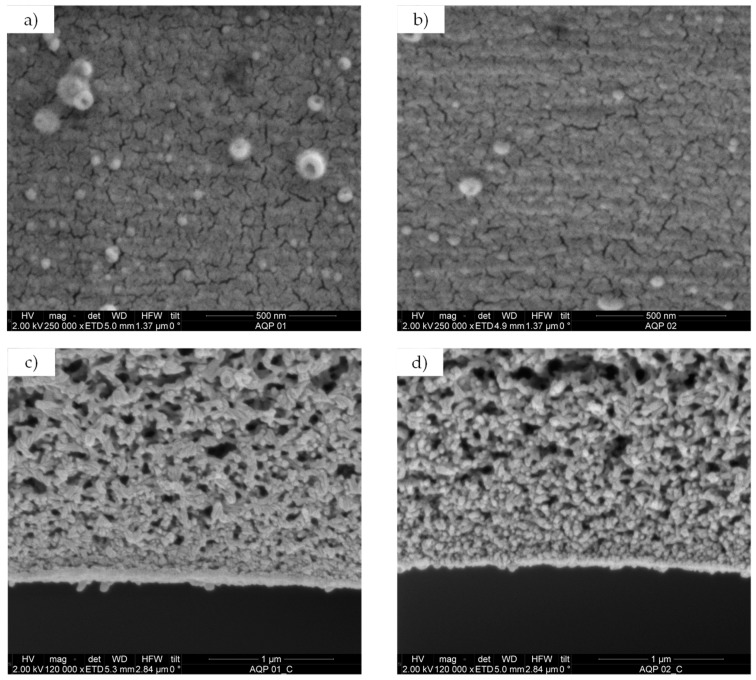
Scanning electron microscopy (SEM) image of the inner surface of (**a**) HF–O, (**b**) HF–C membrane and cross-section of (**c**) HF–O, and (**d**) HF–C membrane.

**Figure 5 membranes-09-00066-f005:**
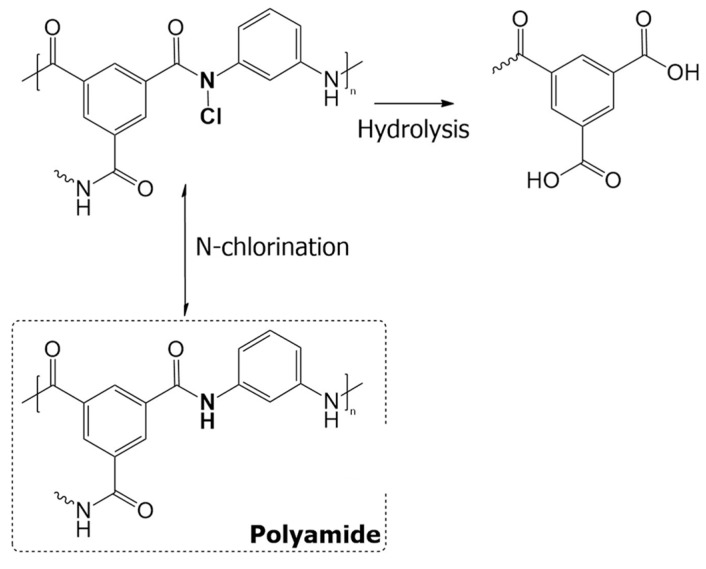
Mechanism of chlorine attack under alkaline conditions with low concentrations of chlorine. Adapted from Verbeke et al. [42].

**Figure 6 membranes-09-00066-f006:**
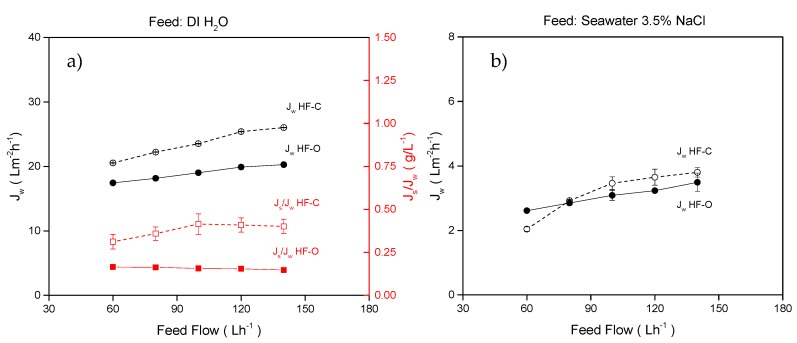
(**a**) J_w_ and specific reverse solute flux (J_s_/J_w_**)** for HF–C and HF–O modules as a function of feed flow rate when DI water was used as a feed solution (FS). (**b**) J_w_ for HF–C and HF–O modules as a function of feed flow rate when 3.5% NaCl solution was used as a FS. Draw flow rate was 25 L·h^−1^, draw concentration was 1 M NaCl and TMP was 0.2 bar. (n = 2).

**Figure 7 membranes-09-00066-f007:**
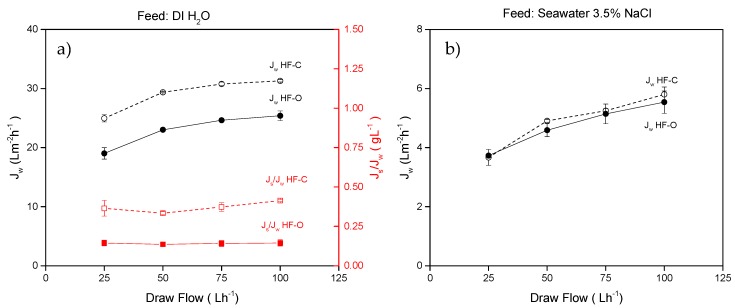
(**a**) J_w_ and J_s_/J_w_ for HF–C and HF–O modules as a function of draw flow rate when DI water was used as an FS; (**b**) J_w_ for HF–C and HF–O modules as a function of draw flow rate when 3.5% NaCl was used as an FS. The feed flow rate was 100 L·h^−1^, the draw concentration was 1 M NaCl, and the TMP was 0.2 bar. (n = 2).

**Figure 8 membranes-09-00066-f008:**
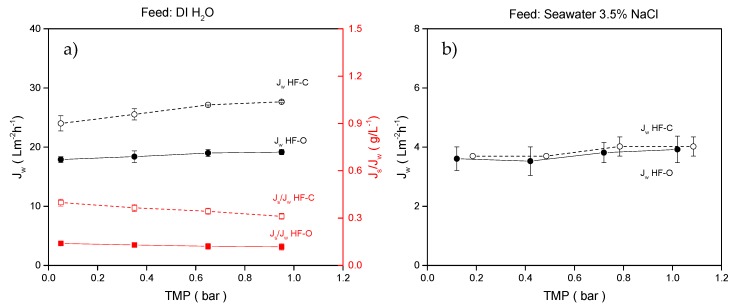
(**a**) J_w_ and J_s_/J_w_ for HF–C and HF–O modules as a function of applied TMP when DI water was used as an FS; (**b**) J_w_ for HF–C and HF–O modules as a function of applied TMP when 3.5% NaCl was used as an FS. Feed flow rate was 100 L·h^−1^, draw flow rate was 25 L·h^−1^, and draw concentration was 1 M NaCl. (n = 2).

**Figure 9 membranes-09-00066-f009:**
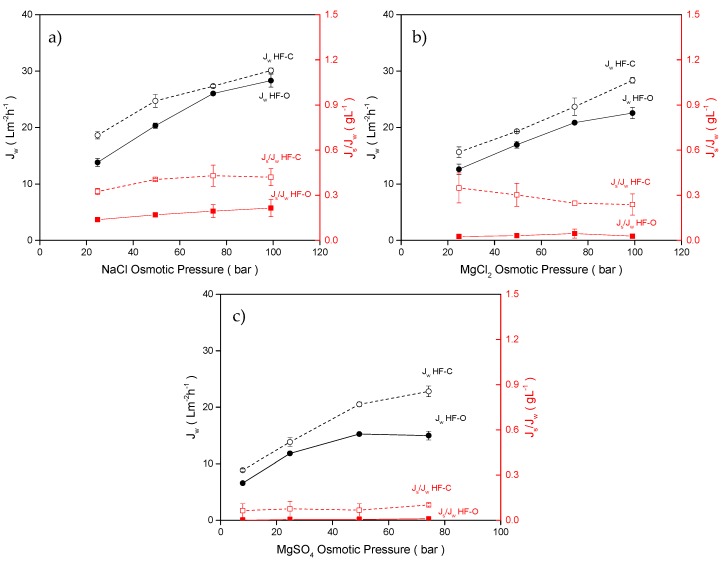
J_w_ and J_s_/J_w_ for HF–C and HF–O modules as a function of the osmotic pressure of (**a**) NaCl DS solution, (**b**) MgCl_2_ DS solution, and (**c**) MgSO_4_ DS solution. All tests were carried out using DI water as an FS. The feed flow rate was 100 L·h^−1^, the draw flow rate was 25 L·h^−1^, and TMP was 0.2 bar (n = 2).

**Figure 10 membranes-09-00066-f010:**
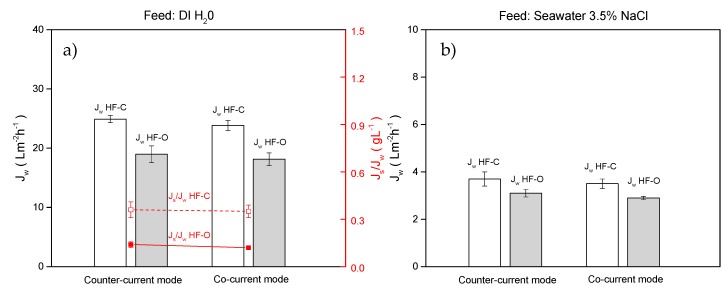
(**a**) J_w_ and J_s_/J_w_ for HF–C and HF–O modules in co-current and counter-current when DI water was used as FS, (**b**) J_w_ for HF–C and HF–O modules in co-current and countercurrent when 3.5% NaCl solution was used as FS. Feed flow rate was 100 L·h^−1^, draw flow rate was 25 L·h^−1^, draw concentration was 1 M NaCl and TMP was 0.2 bar. (n = 2).

**Figure 11 membranes-09-00066-f011:**
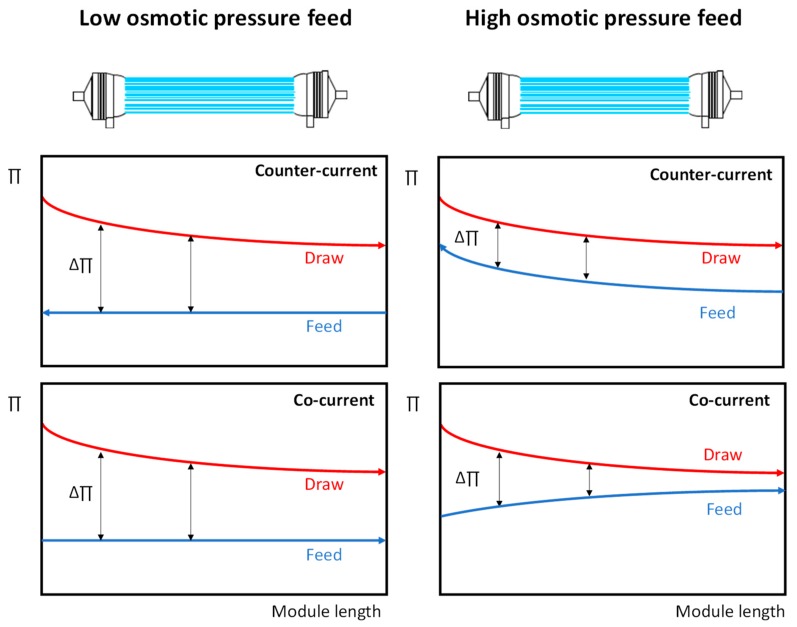
Schematic description of the osmotic pressure development for FS and DS in an HFFO module when operated in a counter-current and co-current mode with low and high osmotic pressure FS.

**Figure 12 membranes-09-00066-f012:**
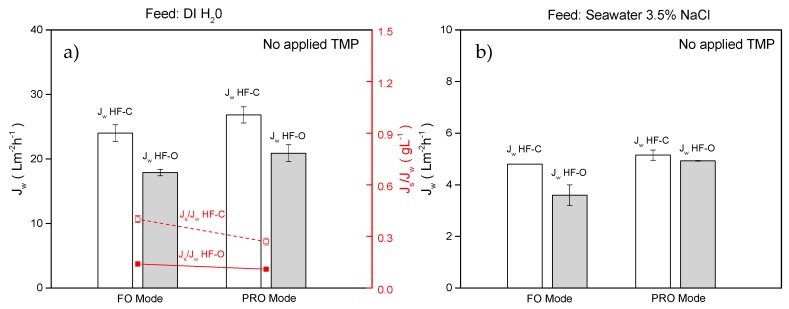
(**a**) J_w_ and J_s_/J_w_ for HF–C and HF–O modules operated in FO and PRO mode when DI water was used as an FS; (**b**) J_w_ for HF–C and HF–O modules operated in FO and PRO mode when 3.5% NaCl was used as an FS. Feed flow rate was 100 L·h^−1^, draw flow rate was 25 L·h^−1^, and draw concentration was 1 M NaCl (n = 2).

**Figure 13 membranes-09-00066-f013:**
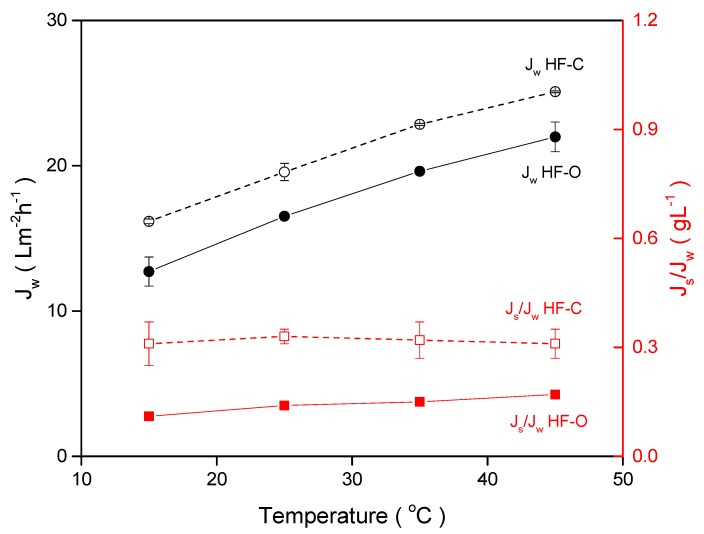
J_w_ and J_s_/J_w_ for HF–C and HF–O modules as a function of the DS and FS temperatures when DI water was used as an FS. The feed flow rate was 100 L·h^−1^, the draw flow rate was 25 L·h^−1^, the draw concentration was 1 M NaCl, and the TMP was 0.2 bar (n = 2).

**Figure 14 membranes-09-00066-f014:**
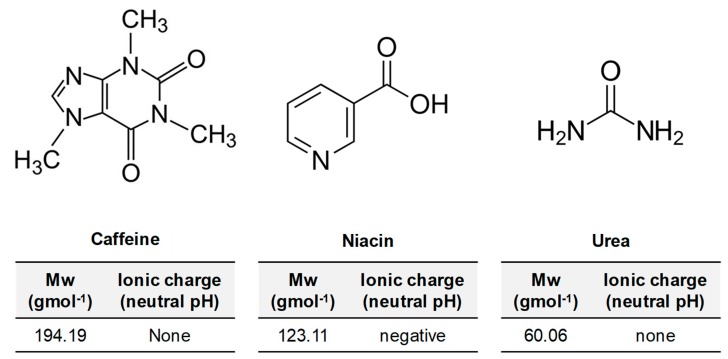
Chemical structures of the model compounds used for the rejection tests. Below the chemical structure, information on the molecular weight (Mw) and molecule charge in a neutral pH solution is described.

**Figure 15 membranes-09-00066-f015:**
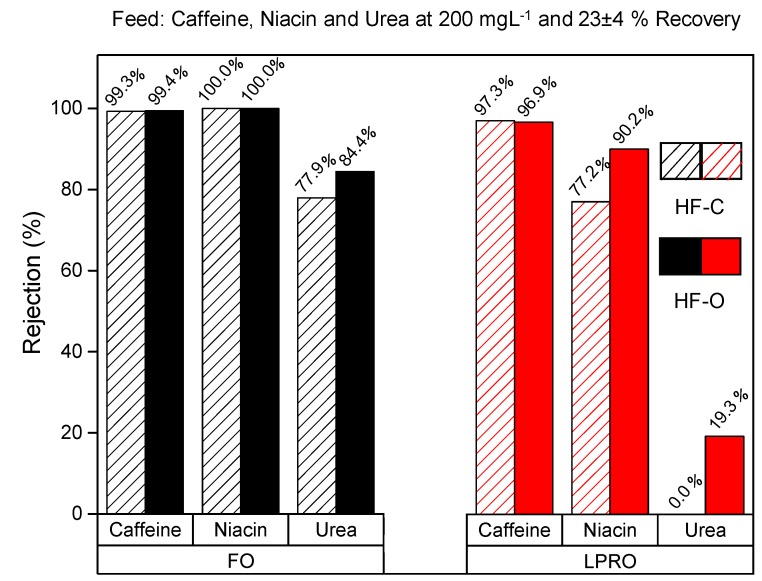
Rejection of caffeine, niacin, and urea for HF–C and HF–O modules in LPRO and FO. Each compound was tested individually with a feed concentration of 200 mg·L^−1^. Feed flow rates were modified for FO (130–140 L·h^−1^) and LPRO (30–60 L·h^−1^) to obtain 23 ± 4% recovery for all the tests. LPRO tests were carried out with a TMP of 2 bar and FO tests were carried out with 0.5 M NaCl as a DS with a flow rate of 16 Lh^−1^ for all the tests.

**Figure 16 membranes-09-00066-f016:**
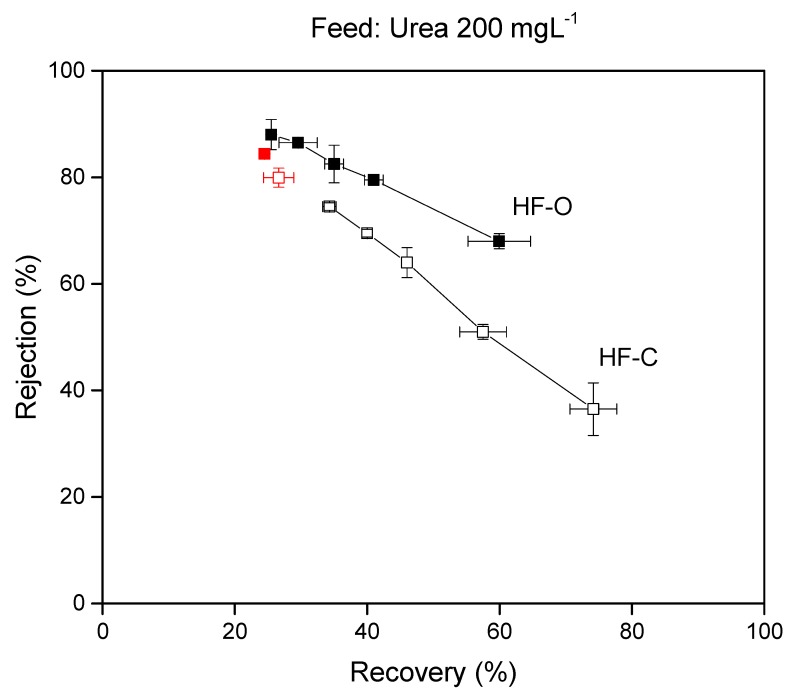
Rejection of urea in FO operation for HF–C and HF–O modules as a function of process recovery. The urea concentration was 200 mg·L^−1^. The feed flow rate for both modules was modified from 60–140 L·h^−1^ to obtain a wide range of process recoveries. The draw flow rate was maintained at 25 L·h^−1^ with a DS of 1M NaCl. Red data points are urea rejections from the FO tests shown in Figure 15, where a different draw flow rate, 16 L·h^−1^, and a different DS concentration, 0.5 NaCl, was used.

**Table 1 membranes-09-00066-t001:** Operational conditions for the standard forward osmosis (FO) tests. During the FO performance tests, only one of the listed testing parameters was modified while other parameters were kept constant. TMP, transmembrane pressure; DI, deionized.

Standard Operational Conditions for the FO Performance Tests							
Feed		Draw		Applied TMP	Temperature	Operation Mode	
Flow	Solution	Flow	Solution	Bar	°C		
100 L·h^−1^	DI H_2_O 3.5% NaCl	25 L·h^−1^	1 M NaCl	0.2	25	FO	Counter-current

**Table 2 membranes-09-00066-t002:** Operational parameters modified during FO tests. PRO, pressure retarded osmosis.

Operational Conditions Modified During the FO Performance Tests					
Feed Flow	Draw Flow	Applied TMP	Operation Mode		Temperature
L·h^−1^	L·h^−1^	Bar	Draw and Feed Flow		°C
60	25	0.1	FO	counter current	15
80	50	0.4	PRO	co-current	25
100	75	0.7			35
120	100	1.1			45
140					

**Table 3 membranes-09-00066-t003:** Characteristics of draw solution (DS) used for the FO tests and for estimation of water permeability coefficient A, solute permeability coefficient B and structural parameter S.

Draw Solutions			
	Concentration	Osmotic Pressure	Diffusivities [45,47,48]
	M	bar	(× 10^−^^9^ m^2^·s^−^^1^)
NaCl	0.5	27.7	1.47
	1	49.5	1.42
	1.5	74.3	1.36
	2	99	1.31
MgCl_2_	0.32	27.7	1.04
	0.58	49.5	1.07
	0.85	74.3	1.09
	1.11	99	1.11
MgSO_4_ *	0.3	7.4	0.49
	0.77	24.7	0.41
	1.54	49.5	0.35
	2.53	74.3	0.30

* Concentrations for MgSO_4_ differ from the other salts due to the solubility limit of MgSO_4_.

**Table 4 membranes-09-00066-t004:** Operational parameters used for the membrane rejection analysis of niacin, caffeine, and urea in FO and low-pressure reverse osmosis (LPRO) processes. The parameters were modified to maintain the same process recoveries between the experiments. HF–C, chlorinated membrane; HF–O, original membrane.

Niacin - Caffeine - Urea Tests *							
Membrane	FO			LPRO		Recovery	
HFFO	Feed Flow	Draw Flow	TMP	Feed Flow	TMP	FO	LPRO
	L·h^−1^	L·h^−1^	bar	L·min^−1^	bar	%	
HF–C	140	16	0.2	1	2	23 ± 4	
HF–O	135	16	0.2	0.5	2	23 ± 4	

* All compounds were tested with 200 mg·L^−1^ of niacin, caffeine or urea in the feed solution.

**Table 5 membranes-09-00066-t005:** Testing conditions used for investigation of the correlation between recovery and urea rejection in the FO process.

HF–C and HF–O Membrane Modules				
Urea FO Test			Recovery	
Feed Flow	Draw Flow	TMP	HF–C	HC-O
L·h^−1^	L·h^−1^	bar	%	%
60	25	0.2	74.1 ± 3.5	59.9 ± 4.7
80	25	0.2	57.5 ± 3.5	41.0 ± 1.4
100	25	0.2	46.0 ± 0.0	35.0 ± 1.4
120	25	0.2	40.0 ± 0.0	29.5 ± 2.8
140	25	0.2	34.2 ± 1.0	25.5 ± 0.7

**Table 6 membranes-09-00066-t006:** Parameters A, B, and S estimated for NaCl, MgCl_2_, and MgSO_4_ draw solute systems. RMSE and R^2^ represent the root mean square error and the coefficient of determination, respectively, for the model. J_w_, water flux; J_s_, reverse solute flux.

Draw Solute NaCl							
Membrane Type	A	B	S	RMSE	R^2^-J_w_	R^2^-J_s_	Remarks
	L·m^−2^·h^−1^·bar^−1^	L·m^−2^·h^−1^	mm	-	-	-	
HF–O	1.56	0.24	0.15	0.54	0.99	0.76	-
HF–C	4.04	1.29	0.26	0.77	0.98	0.79	Out of range
	2.46	0.80	0.18	1.22	0.92	0.95	Fixed A = 2.46
**Draw Solute MgCl_2_**							
HF–O	1.86	0.07	0.16	0.40	0.99	0.02	-
HF–C	2.74	0.59	0.15	0.74	0.97	0.64	-
**Draw Solute MgSO_4_**							
HF–O	1.68	0.01	0.12	0.43	0.97	0.11	-
HF–C	2.04	0.10	0.08	0.21	1.00	0.78	Out of range
	2.46	0.12	0.09	0.44	0.98	0.50	Fixed A = 2.46

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
