# Peer review of "Role of Operating Conditions in a Pilot Scale Investigation of Hollow Fiber Forward Osmosis Membrane Modules"

_membranes, 2019, doi:10.3390/membranes9060066_

Round 1

Reviewer 1 Report

This is very interesting study on the aquaporin based hollow fiber forward osmosis (HFFO) module performance evaluation at different operating conditions before and after modification. This is extensive study with clear experimental design for FO test. However, there are some points need to be improved for publication.

1)     Abstract: Line 15 – “there is no clear systematization of testing conditions at pilot scale for FO membrane modules” – There are several papers studied in the recent literature, Line 16 – “two types of HFFO”, here type is unclear term, in fact HFFO was modified or not (original). Line 22-27 – These points seem not significant findings. Please re-write the abstract, which is most important part of this manuscript.

2)     Line 63-66: This statement needs references to support it.

3)     Line 112-113, 130: Expose the membrane to NaOCl – How much do you guarantee this method works for aquaporin HFFO membrane?

4)     Line 142-143: Experimental design is somewhat confusing. A mini-modules are used for only zeta potential measurement? It should be clearly mentioned.

5)     Section 2.2.3 can be located in supplementary information.

6)     Line 265-266: Why the FO test was running only for 45 min? Table 1: Why those feed and draw flow rates are chosen?

7)     Figure 4: What are meaning of different symbols?

8)     Line 388: Here, thickness of PA layer is reduced. It means that membrane were scratched out? In this case, the rejection and flux should be affected. I found the water flux is changed but couldn’t find the RSF.

9)     Figure 7: Experimental condition is not clear. Please provide more details in the caption.

10) Section 3.2.3: TMP applied in this study is not enough to see the different performances. How did you expect in this study?

11) Figure 10: It just looks the differences were made only due to the osmotic pressure, not depending on draw solute types.

12) Conclusion: This is too long. The conclusion is not summary of the results obtained from each section.           

Author Response

Attached Word file with answers.

Reviewer 2 Report

In this manuscript, performance of two types of hollow fiber forward osmosis membrane modules have been investigated at different operating conditions. Although plenty of data were presented, there are still some shortcomings in the paper. I believe the paper may be accepted for publication after carefully addressing the following points.

The title of your paper is" Performance analysis of hollow fiber forward osmosis

membrane modules",but in your article, you discussed the impact of operating

conditions on test results.The title does not match the content of the article. The scope of the title is too wide.

In your introduction, you didn’t explain why sodium hypochlorite should be used for post-treatment.To add a set of data to make the analysis more reliable or to modify the hollow fiber membrane ? Although the electronic formula for sodium hypochlorite is NaOCl, it is better to write its chemical formula (NaClO) in the paper.

.The conclusion is too complicated. You'd better simplify it.

4. .In this paper, the analysis of different results caused by different draw solutions is not thorough enough, which can be analyzed from the perspective of ion size.Hui Kang etal.Applied Surface Science,2019,465:1103-1106 may give you some assistance. Some more papers for forward osmosis could be cited in Introduction to show the novelty in this paper.

5. There are many grammar and tense problems, such as :

(1)Page 2,line 46, "hollow fiber configuration has proven to have multiple advantages compared to other form factors" should be wrote"hollow fiber configuration has been proven to have multiple advantages compared to other form factors ".

(2)Page 2, line 61,"To date, assessment of FO membranes have not been standardized and multiple examples of the testing conditions and testing setups configurations have been reported "should be wrote "To date, assessments of FO membranes have not been standardized and multiple examples of the testing conditions and testing setups configurations have been reported".

(3)Page 4, line 161,"In addition, water molecules, moving from feed to draw, dilutes solute concentration on the porous substrate surface, creating a reduction in concentration between draw bulk solution and support layer surface (dilutive ECP)"should be written "In addition, water molecules, moving from feed to draw, dilute solute concentration on the porous substrate surface, creating a reduction in concentration between draw bulk solution and support layer surface (dilutive ECP)".

(4)Page 5, line 191,"for feed and draw side,  an average cross-flow velocity (v̅F and v̅F) are defined according to equation 3 and 4" should be written "for feed and draw side,  an average cross-flow velocity (v̅F and v̅D) are defined according to equation 3 and 4"and "where RFO represent the membrane rejection in FO process"should be written "where RFO represents the membrane rejection in FO process"

(5)Page 9, line 311 and 315,"where CD,out and Cpermeate represents the concentrations of the target compound in draw outflow and permeate stream" should be written"where CD,out and Cpermeate represent the concentrations of the target compound in draw outflow and permeate stream"

(6)Page 13, line 409"the cleavage might result in the reduction of the cross-linking and an increase of the number of carboxylic groups in the polyamide structure, which enhances the hydrophilicity of the selective layer. "should be written "the cleavage might result in the reduction of the cross-linking and an increase of the number of hydroxy groups in the polyamide structure, which enhances the hydrophilicity of the selective layer. "

(7)Page 15, line 469"For DI water as FS, HF-O module exhibited Jw increase from 17.4 of 20.2 Lm-2h-1 (about 16.2%) in comparison to HF-C modules that showed Jw increase from 20.5 to 26 Lm-2h-1 (about 26.7%) by varying feed flow rate in the tested range "should be written "For DI water as FS, HF-O module exhibited Jw increase from 17.4 to 20.2 Lm-2h-1 (about 16.2%) in comparison to HF-C modules that showed Jw increase from 20.5 to 26 Lm-2h-1 (about 26.7%) by varying feed flow rate in the tested range "

(8)Page 22, line 723 and 726 "Seker et al. highlight the importance of the temperature for viscous FS and describe how the efficiency of the FO process can be enhanced by an increase of the FS temperature, which reduced its viscosity while keeping DS cold to reduce draw solute diffusion to the FS. In addition, Xie et al. demonstrates that the increase in Jw with temperature can come with the negative side effect of the decrease in solute rejection of trace organic compounds due to the increase in solute diffusivity"should be wrote"Seker et al. highlighted the importance of the temperature for viscous FS and describe how the efficiency of the FO process can be enhanced by an increase of the FS temperature, which reduced its viscosity while keeping DS cold to reduce draw solute diffusion to the FS. In addition, Xie et al. demonstrated that the increase in Jw with temperature can come with the negative side effect of the decrease in solute rejection of trace organic compounds due to the increase in solute diffusivity".

Author Response

Attached Word file with answers.

Reviewer 3 Report

The paper systematically studied the performances of the FO hollow fiber membranes received from Aquaporin A/S. To compare the membrane performance, they produced the looser FO membrane using NaOCl. Although the configuration of hollow fibers in the modules is very critical in the FO performance (https://doi.org/10.1002/app.46110), the detailed explanation for ECP is not included because it is very complicated. Nevertheless, the results using biomimetic FO hollow fiber module could give some merits in the FO scientific field. Therefore, it would be proper to be published in Membranes.

Sincerely

Author Response

Response to Reviewer 3 comments

The paper systematically studied the performances of the FO hollow fiber membranes received from Aquaporin A/S. To compare the membrane performance, they produced the looser FO membrane using NaOCl. Although the configuration of hollow fibers in the modules is very critical in the FO performance (https://doi.org/10.1002/app.46110), the detailed explanation for ECP is not included because it is very complicated. Nevertheless, the results using biomimetic FO hollow fiber module could give some merits in the FO scientific field. Therefore, it would be proper to be published in Membranes.

We would like to thank the reviewer for the thoughtful reading of the paper and helpful observations provided. The comment of the reviewer is exact; the configuration of the hollow fibers in the module is very critical to the FO performance. We explained ECP first in Line 410 and further mentioned it for other sections, nonetheless it is indeed true that we have not explained it in detail as we have focused more in ICP due to its relevant role in membrane performance. We have added a small explanation of the importance of these factors, with their references, in the Introduction section.

Reviewer 4 Report

In this study, the authors investigated the pilot scale FO membrane performance at different operation conditions. The manuscript has well written and the reviewer only has a few comments, which need to be further strengthened and explained.

(1)  The ‘pilot scale’ should be added in the title.

(2)  NaClO is normally used for membrane cleaning. Here the authors emphasized that NaClO was used for FO modification. Is that a really modification process of FO membrane, or just cleaning the virgin FO membrane which may contain impurity substances during manufacturing process. The author should provide evidence for it, such as FTIR analysis instead of zeta potential and SEM images (those are not direct evidences).

(3)  For the FO operation, the crossflow velocity should be presented instead of flow rate.

(4)  The authors should give statistical calculation of p-value of the compared data.

Author Response

Attached Word file with answers.

Round 2

Reviewer 1 Report

Well revised and manuscript has been improved to be published. However, figure scale (y-axis) of Fig. 3, 6b, 7b, 8b and 13 should be changed to show the patten clearly.

Author Response

We thank the reviewer for the comment and time put in this second round of revision. As suggested, Figure 3 and Figure 13 were modified and the range of the y-axis was reduced for a better understating of the patterns of the membrane performance. For Figures 6-8, we modified the y-axis range to maintain the same y-axis size between the graphs.

Reviewer 2 Report

The experimental data are very sufficient. The effect of different operating conditions on the performance of the membranes is also analyzed in detail, which can be used as a guide for the testing of hollow fiber membranes in the future.

Author Response

We greatly appreciate the reviewer for the comments and time put in this second round of revision and we are happy to meet the expectations.